# Phenotypic Diversity Analysis and Superior Family Selection of Industrial Raw Material Forest Species-*Pinus yunnanensis* Franch

**Zirui Liu** [1,2]**, Chengjie Gao** [1]**, Jin Li** [1]**, Yingchun Miao** [1] **and Kai Cui** [1,*]

[1]  State Key Laboratory of Tree Genetics and Breeding, Institute of Highland Forest Science, Chinese Academy of Forestry, Kunming 650233, China; lzramy@sina.com (Z.L.); gcj1986113@163.com (C.G.); lijin@caf.ac.cn (J.L.); myczsh@163.com (Y.M.)
[2]  College of Forestry, Nanjing Forestry University, Nanjing 210037, China
[*]  Correspondence: cuikai@caf.ac.cn; Tel.: +86-871-6386-0026; Fax: +86-871-6386-0027

**Abstract:** *Pinus yunnanensis* Franch is a major forest species in southwest China as a source of timber and industrial raw materials. The genetic quality of the species is declining and the differentiation of offspring is strong as affected by environmental change and improper management measures. To assess the phenotypic diversity of natural populations, the evaluation of twelve phenotypic traits in nine populations from its whole distribution was performed. Studies revealed plentiful phenotypic variations within and among populations. The phenotypic variation within the population was 4.03%, and was lower than that among populations (21.04%), indicating that the phenotypic variation among populations was the main source. The mean differentiation coefficient was 91.23%, and the mean coefficient of variation of twelve traits was 28.27%, ranging from 14.18% (length of needles) to 70.11% (height under the branches). No significant correlation between plant height and environmental factors was found. Mean annual temperature, mean temperature of the driest quarter, mean temperature of the wettest quarter, and latitude were significantly correlated to diameter breast height, respectively. Temperature is the most important factor affecting the diameter of breast height. Three principal components that represent plant shape, needle, and lateral branch trait, respectively, were obtained while the cumulative contribution rate reached 74.40%. According to the unweighted pair-group method with arithmetic means (UPGMA) cluster analysis, nine populations were divided into three clusters. However, populations were not clustered strictly according to geographic distance, implying that there is a discontinuity in the variation of phenotypic traits. Compared with other populations, the Lufeng population contains obvious advantages in plant height, diameter breast height, crown diameter, and needle length and width, whereas the Yongren population has the worst performance in plant height, crown diameter, and the number of lateral branches. Moreover, for selecting superior families, both the comprehensive scoring method and principal component analysis were combined. By comparing trait values from 258 families, eleven superior families with an actual gain of each trait ranging from 0.02% to 32.23% were successfully screened out. This study provides a certain reference significance for the breeding of improved varieties and plantation management of *P. yunnanensis*.

**Keywords:** *Pinus yunnanensis* Franch; natural population; phenotypic trait; superior tree; artificial selection

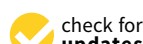



## 1. Introduction

*Pinus yunnanensis* is an aiphyllium of Pinaceae, which is the main timber and afforestation species in southwest China [1]. It is distributed in eastern Tibet, western and southwestern Sichuan, Yunnan, western and southwestern Guizhou, and northwestern Guangxi. The distribution range is from 23–30° N to 96–108° E [2]. The existing forest area is over 6 million hectares, and the forest stock volume is 28.67 million m³ [3]. It can not only grow on the limestone-developed red soil but also on the barren gravel ground or severely washed barren hills where other species cannot grow. It is a pioneer tree species for afforestation in barren hills and an important native species for controlling soil erosion in

the distribution area [4]. It has the characteristics of light-demanding, resistance to drought, and barren and strong adaptability. It can also be used for high-value solid wood products such as furniture and paper. In addition, there is rich pine resin in *P. yunnanensis*, and the existing storage capacity is about $1.2 \times 10^6$ kg [5], which ranks at the forefront of the world. The output value of industries involved in rosin and turpentine series products, which are the main components of pine resin, accounts for 1/10 of the total output value of the Chinese national economy. Colophony is widely used in papermaking, printing inks, coatings, surfactants, and food additives industries [6]. In addition, turpentine oil is also widely used in pesticides [7,8] and biomass fuel [9].

Phenotypic diversity comprehensively reflects genetic diversity and environmental diversity. It is often expressed as phenotypic variation produced by different populations to adapt to environmental conditions in different distribution areas. Phenotypic variation is of great significance in adaptation and evolution [10,11]. Phenotypic trait analysis is a relatively intuitive and effective method to study plant genetic diversity. Studying plant phenotypic traits can not only understand the mechanism of plant genotype and environmental interaction but also dig out specific traits with higher economic and ecological value. It has an important theoretical and practical sense for promoting the innovation, creation, and efficient use of plant germplasm resources [12]. Conducted a study on the phenotypic diversity of needle characteristics of *P. yunnanensis* populations distributed at different altitudes, which proved that the needle characteristics have extremely rich variation among and within populations, and concluded that there is no significant correlation between the needle traits and altitude [13]. The phenotypic traits selected were mainly interested in forestry production, and were also easily accessible, and guided for forestry production. Through the study of the variation of needle and cone traits of the natural population, it was proved that the variation of needle and cone traits mainly came from within the population and that the variation of needle traits was related to geographic and climatic factors. The needle traits' phenotypic variations between the different types of trunks were analyzed, and it was known that the average values of the needle traits in the canopy layer were greater than that for the regeneration layer in both the straight population and twisted population. However, the needle shape index had little difference between the canopy and regeneration layer in both straight and twisted populations. At present, the reported studies relating to phenotypic variation of *P. yunnanensis* mostly focus on the characteristics of needles and cones, so more diverse traits need to be explored. In addition, the scope of previous research was regional, while phenotypic diversity of the whole natural distribution area has not been reported, which is not conducive to the development of breeding and selection of *P. yunnanensis*.

Superior family and individual selection can not only improve the genetic structure of plantation but also promote the growth of young forests. On the other hand, seedling selection can shorten the breeding cycle and maintain the excellent genetic characteristics of parents by asexual reproduction [14,15]. The heredity and variation of populations, families, or clones are the basis and important materials of forest breeding. Understanding the genetic variation parameters of different breeding populations is very important for the selection of superior materials [16]. The phenotypic variation coefficient can clearly reflect the variation of various indicators among different populations, and a larger phenotypic variation coefficient is conducive to evaluation selection for superior populations [17]. Gao et al. (1984) adopted the comparative tree comparison method to select the superior trees of *P. yunnanensis*, based on the growth rate, the straightness of the trunk, and the degree of wood grain twisting, which developed an index system for the selection of the superior trees. The predecessors established a tree selection system by evaluating qualitative traits. However, we believe that the inability to quantify these qualitative indicators is a big drawback. Therefore, all quantitative traits were selected in our study.

The heredity and variation of populations, families, or clones are the basis and important materials for tree breeding [16]. Understanding the genetic variation parameters of different breeding populations is very important for the selection of excellent materials. The

phenotypic variation coefficient can clearly reflect the variation of various indicators among different families. The greater the phenotypic variation coefficient, the more conducive to the evaluation and selection of excellent families [17]. Provenance and family selection is one of the important means of tree genetic improvement. A large number of studies have proved that good improvement results can be achieved by provenance and family selection [18–21].

In this study, twelve phenotypic traits in nine natural populations were used for the research in order to explore the characteristics of phenotypic variation and the relationship between phenotypic traits and spatial arrangement and ecological factors. Moreover, the superior family selection was performed combined with the weighted scoring and principal component analysis. This work is beneficial to provide a theoretical basis for provenance division, collection and protection of germplasm resources, and selection and breeding of superior varieties.

## 2. Materials and Methods

### 2.1. Site Description and Stand Characteristics

Nine provenances distributed in four provinces of Yunnan, Sichuan, Guizhou, and Tibet were surveyed and selected, including Ceheng County (CH), Xichang City (XC), Chayu County (CY), Shuangbai County (SB), Lufeng County (LF), Yongren County (YR), Huize County (HZ), Tianchi County (TC), and Xinping County (XP). Each sampling point was set as a population. According to plant height, plant shape, and health degree, about 30 superior trees with more than 30 m spacing between individual plants were selected for each population. Thus, all populations can basically represent the overall distribution characteristics of the natural population of *P. yunnanensis*. Seeds were collected in four directions of each tree crown [22], and the provenance and family experimental forest was built in Lufeng County, Yunnan Province in 2016. A total of 258 families from nine provenances participated in the test. The random block design composed of a single plant plot with 30 repetitions was adopted, and the planting spacing of afforestation is 2 m × 3 m. The geographic locations of the nine survey sites are shown (Figure 1). The specific geographic distribution and environmental climate factors of the survey sites are shown (Table 1).

In order to eliminate environmental differences, the horizontal banded land preparation before afforestation was used. For provenance and family experimental forest, each provenance is randomly arranged within each block, and each family is randomly arranged within the provenance.

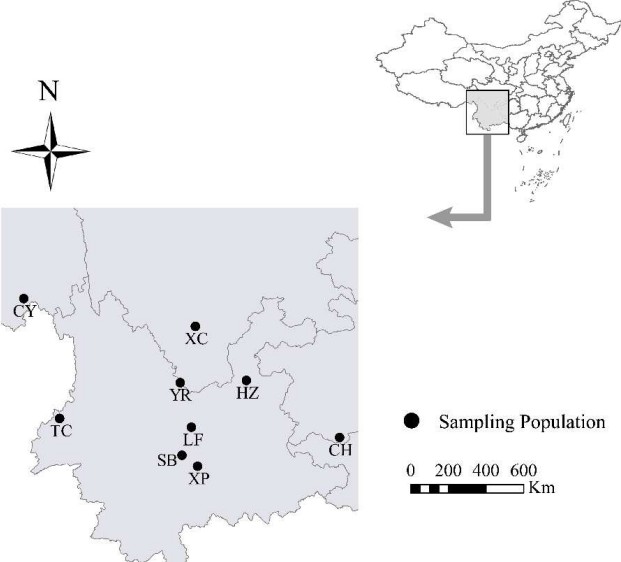

**Figure 1.** Locations of nine sampling populations of *P. yunnanensis*.

**Table 1.** Geographical locations and main climatic conditions for the nine *P. yunnanensis* populations sampled.

| Population | Individual Samples | LON (°E) | LAT (°N) | EL (m) | MAP (mm) | MAT (°C) | WP (mm) | WT (°C) | DP (mm) | DT (°C) |
|---|---|---|---|---|---|---|---|---|---|---|
| Ceheng (CH) | 90 | 105.93 | 24.85 | 800 | 1256 | 17.9 | 705 | 24.6 | 29.5 | 9.9 |
| Xichang (XC) | 87 | 102.01 | 27.87 | 2610 | 918 | 7.8 | 518 | 14.0 | 22 | 0.3 |
| Shuangbai (SB) | 84 | 101.65 | 24.37 | 1655 | 1004 | 20.7 | 567 | 24.9 | 29.4 | 16.8 |
| Chayu (CY) | 84 | 97.35 | 28.62 | 2050 | 791 | 4.9 | 524 | 11.5 | 42 | −0.9 |
| Huize (HZ) | 90 | 103.40 | 26.40 | 2320 | 928 | 10.5 | 516 | 16.4 | 28 | 3.7 |
| Lufeng (LF) | 87 | 101.90 | 25.13 | 1925 | 817 | 15.9 | 450 | 20.9 | 39 | 11.4 |
| Tianchi (TC) | 90 | 98.32 | 25.37 | 2125 | 1442 | 12.6 | 743 | 17.4 | 76 | 6.8 |
| Yongren (YR) | 90 | 101.60 | 26.34 | 2055 | 825 | 13.9 | 467 | 19.0 | 29 | 7.9 |
| Xinping (XP) | 72 | 102.07 | 24.07 | 1600 | 974 | 18.2 | 550 | 22.5 | 27.2 | 12.3 |

LON, Longitude; LAT, Latitude; EL, Elevation; MAP, mean annual precipitation; MAT, mean annual temperature; WP, mean precipitation of wettest quarter; WT, mean temperature of the wettest quarter; DP, mean precipitation of driest quarter; DT, mean temperature of the driest quarter.

## 2.2. Measurement of Phenotypic Traits

Phenotypic determination was performed on the five-year-old seedlings. Twelve phenotypic traits that are genetically relatively stable, easy to obtain, and measure were selected, namely plant height (PH), diameter breast height (DBH), long crown diameter (LCD), short crown diameter (SCD), height under the branches (HUB), length of the main branch of the year (LMB), number of lateral branches (NLB), number of lateral branches of the year (NLBY), length of needles (LN), the width of needles (WN), length of leaf sheath (LLS), and needles fascicles width (NFW). The plant height and height under the branches were measured with a tree altimeter (accuracy of 0.1 m), and the diameter breast height was measured with a vernier calliper (accuracy of 0.01 mm). In addition, the crown diameter was measured with a tower ruler (accuracy of 0.1 m), which was the mean value along the slope and contour. Long crown diameter and short crown diameter were referred to as the maximum and minimum tree crown diameter, respectively. Every tree in the experimental forest was investigated. The related traits of crown and diameter of each tree were measured three times, respectively, and the measurement method and trait type were shown (Table S1).

## 2.3. Standardized Data

Among the twelve traits, only the number of lateral branches and the number of lateral branches of the year were measured in a strip, and the other traits are measured in centimeters. Therefore, in order to facilitate the scoring, correlation analysis, and principal component analysis of the number of lateral branches and the number of lateral branches of the year, the multi-objective decision-making one-dimensional comparison method was used to standardize the indices. The conversion formula is as follows:

$$U = 1 - 0.9 \times (T - T_{min})/(T_{max} - T_{min}) \tag{1}$$

$T$ is the measured value of each individual, and $T_{max}$ and $T_{min}$ are the measured maximum and minimum values, respectively.

## 2.4. Data Analysis

Nested variance analysis was used to study the significance of differences in phenotypic traits among and within populations. The linear model is $y_{ijn} = \mu + \alpha_i + \beta_{j(i)} + \varepsilon_{ijn}$, where $y_{ijn}$ is the $n$ observation value of the $j$ individual of the $i$ population, $\mu$ means overall average, $\alpha_i$ is the fixed effect value of the $i$ population, $\beta_{j(i)}$ is the random effective value of the $j$ individual in the $i$ population, and $\varepsilon_{ijn}$ is the experimental error of the $ijn$ observation value. Duncan's multiple comparisons and difference test was performed, and the average value ($\overline{X}$), standard deviation (*SD*), phenotypic differentiation coefficient

($V_{st}$), and coefficient of variation (*CV*) for each trait of nine populations were calculated, respectively [23].

$$V_{st} = \delta^2_{t/s} \left( \delta^2_{t/s} + \delta^2_s \right), \; CV = SD/\overline{X} \tag{2}$$

$\delta^2_{t/s}$ is the variance between populations, $\delta^2_s$ is the variance within the population [24–26]. Based on the data of twelve traits, the unweighted group average method (UPGMA) cluster analysis was performed on nine populations [27,28]. The Pearson coefficient was used to analyze the correlation between phenotypic traits and spatial arrangement and ecological factors. Among them, the nested variance analysis, multiple comparisons, variance analysis, correlation analysis, principal component analysis, and cluster analysis are all analyzed and plotted with SPSS 20.0 and R software packages.

### 2.5. Superior Family Selection Method

In this study, a comprehensive scoring method and principal component analysis method were used to select the superior family.

Comprehensive scoring method: each trait was scored based on the analysis of the mean, range, standard deviation, and basic value (mean + standard deviation). One-tenth of the range of each trait value was taken as the scoring grade. For the number of lateral branches and number of lateral branches of the year, the value is between 0–1 after normalization, thus the scoring grade is divided into 0 to 1 and the twelve traits' scores ranged from 1 to 10. The total score of each family was calculated and a *t*-test was used to check whether the data conformed to the normal distribution. The minimum score line was calculated according to the following formula:

$$G = G_0 - t_{0.05} \delta / \sqrt{n} \tag{3}$$

where $G_0$ is the average score, $t_{0.05}$ is the test value, $\delta$ is the standard deviation, and *n* is the number of families. A heat map was applied to indicate 1–10 points with different colors, and 258 families were clustered according to the score value. Finally, based on the color and clustering results of the heat map, the superior families with higher average scores for each trait were evaluated with a selection rate of 10%. According to the formula (3): (among them, $G_0$ is the mean value of 70, $t_{0.05}$ is the test value of 1.65, $\delta$ is the standard deviation of 9.41, and *n* is the number of single plants 258), it was calculated that the lower limit of the average score is 69 points, and a family with a score higher than 69 is determined as the primary selected family.

Principal component analysis: The comprehensive weight of each principal component was calculated (the variance contribution rate corresponding to each principal component was divided by the accumulative contribution rate). Excellent families were screened with a specific selection rate.

## 3. Results

### 3.1. Phenotypic Variation among and within Populations

The variation analyses of twelve phenotypic traits were divided into two levels: among populations and within a population. Except for short crown diameter, number of lateral branches, number of lateral branches of the year, width of needles and needles fascicles width with no significant differences within a population, the significant ($p < 0.05$) differences among and within populations of other tested traits were shown (Table 2). By analyzing $\overline{X}$, *SD* and multiple comparisons of phenotypic traits (Figure 2), there were significant differences in phenotypic traits among populations. The width of the needles in the CH population was the largest, while the CY population had the largest value of height under the branches, the number of lateral branches, and the number of lateral branches of the year. The value of plant height, diameter breast height, long crown diameter, short crown diameter, length of the main branch of the year, length of needles, the width of needles, length of leaf sheath, and needles fascicles width were the largest in the LF population.

**Table 2.** Variance analysis and phenotypic differentiation coefficients among the *P. yunnanensis* populations sampled.

| Traits | F Value | | Proportion of Variance Components (%) | | | Phenotypic Differentiation Coefficients% |
|---|---|---|---|---|---|---|
| | Among Populations | Within Population | Among Populations | Within Population | Random Error | |
| PH | 19.92 ** | 2.12 ** | 115.01 | 22.49 | 6.15 | 90.39 |
| DBH | 20.91 ** | 1.90 ** | 14.61 | 2.19 | 62.54 | 91.69 |
| LCD | 11.01 ** | 1.69 * | 42.45 | 9.80 | 2.83 | 86.67 |
| SCD | 7.23 ** | 1.38 | 28.06 | 6.06 | 2.45 | 83.97 |
| HUB | 5.40 ** | 1.67 * | 3.60 | 1.76 | 65.73 | 76.39 |
| LMB | 10.04 ** | 1.64 * | 21.79 | 4.75 | 3.47 | 85.98 |
| NLB | 35.11 ** | 1.40 | 5.63 | 0.23 | 14.13 | 96.18 |
| NLBY | 54.59 ** | 0.92 | 3.04 | 0.00 | 4.75 | 98.34 |
| LN | 76.91 ** | 2.12 ** | 12.51 | 0.61 | 14.12 | 97.32 |
| WN | 34.38 ** | 1.26 | 0.02 | 0.01 | 0.04 | 96.45 |
| LLS | 74.72 ** | 2.68 ** | 5.71 | 0.44 | 6.73 | 96.54 |
| NFW | 22.38 ** | 1.22 | 0.05 | 0.02 | 0.22 | 94.82 |
| Mean | | | 21.04 | 4.03 | 15.26 | 91.23 |

* $p < 0.05$; ** $p < 0.01$; Phenotypic trait abbreviations are shown in Figure 2.

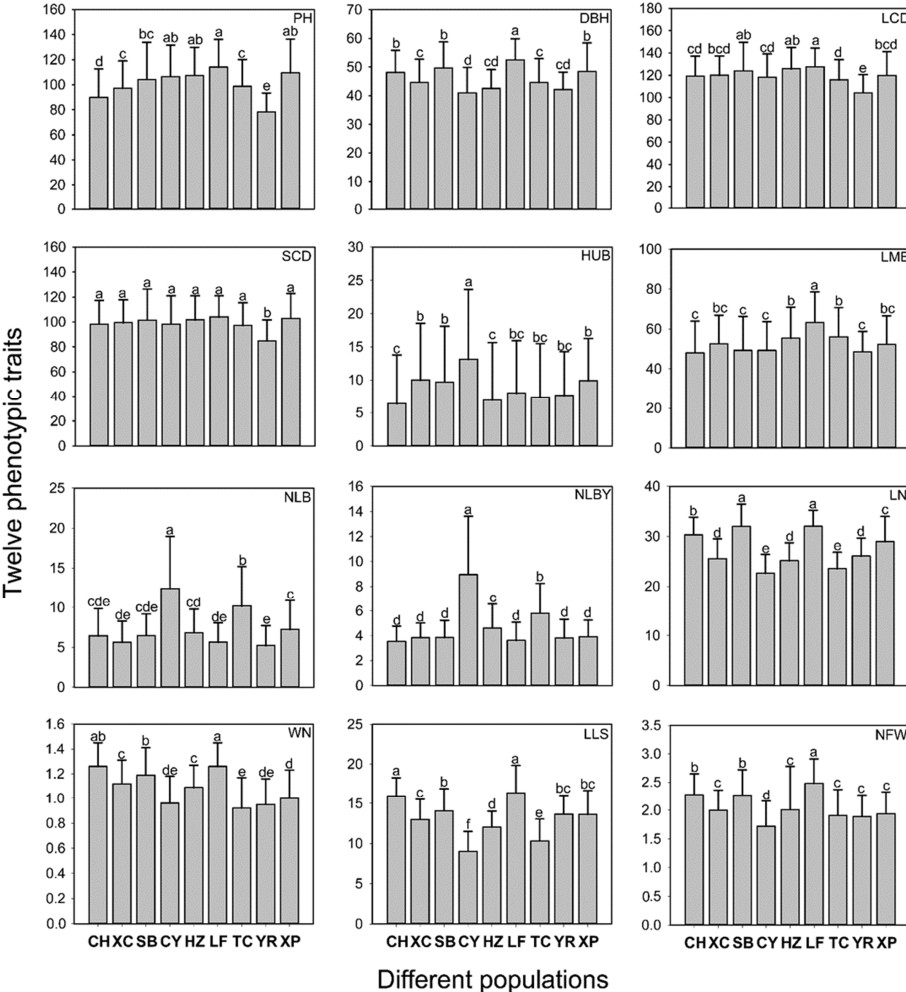

**Figure 2.** Phenotype traits and multiple comparisons of nine *P. yunnanensis* populations. Different letters in each graph indicate significant differences at $p < 0.05$ based on Duncan's multiple comparisons and difference test. Error bar represents SD. PH, plant height; DBH, diameter breast height; LCD, long crown diameter; SCD, short crown diameter; HUB, height under the branches; LMB, length of the main branch of the year; NLB, number of lateral branches; NLBY, number of lateral branches of the year; LN, length of needles; WN, width of needles; LLS, length of leaf sheath; NFW, needles fascicles width. Population abbreviations and phenotypic trait abbreviations are shown in Table 1.

### 3.2. Phenotypic Differentiation among Populations

The proportion of variance components and phenotypic differentiation coefficients of twelve phenotypic traits among and within populations were obtained by the nested variance analysis (Table 2). The results showed that the average proportions of variance components are 21.04% and 4.03%, respectively. The phenotypic differentiation coefficients between populations ranged from 76.39% to 98.34%. Among them, the largest phenotypic differentiation coefficient is the number of lateral branches of the year, and the smallest one is the height under the branches, indicating that the lateral branches are highly differentiated among populations while the height under the branches is relatively stable. The phenotypic differentiation coefficients of all phenotypes are more than 50%, demonstrating that the diversity of all traits among populations is more abundant than that within a population. In short, the average value of the phenotypic differentiation coefficient of each trait is 91.23%, that is, the contribution rate among populations variation in phenotypic variation is 91.23%, whereas the contribution rate within the population is 8.77%. The variation within the population is smaller than that among the populations. The phenotypic diversity of natural populations mainly comes from the variation among populations.

### 3.3. Variation Characteristics of Phenotypic Traits

The coefficient of variation is used to compare the degree of dispersion of phenotypic traits. The larger the coefficient of variation, the higher the dispersion of representative traits, and the greater the degree of phenotypic variability. Analysis of the variance components of nine natural populations (Table 3) showed that the average variance coefficient of twelve phenotypic traits is 28.28%, with a range from 14.17% to 70.18%. Among them, the largest variance coefficient is the height under the branches (70.18%), followed by the number of lateral branches (48.00%) and the number of lateral branches of the year (39.25%), and the length of needles (14.17%) is the smallest. The average variance coefficient of the phenotypic traits of the nine populations ranges from 25.04% to 32.70%. Among them, the CY population has the richest phenotypic diversity, while LF and YR populations have relatively low phenotypic diversity.

**Table 3.** Characteristic of phenotypic traits in populations of *P. yunnanensis* revealed by variance components.

| Traits | Population | | | | | | | | | Mean |
|--------|------|------|------|------|------|------|------|------|------|------|
| | **CH** | **XC** | **SB** | **CY** | **HZ** | **LF** | **TC** | **YR** | **XP** | |
| PH | 25.19 | 22.66 | 28.66 | 23.76 | 21.10 | 19.48 | 21.78 | 19.35 | 24.73 | 22.97 |
| DBH | 16.13 | 18.07 | 18.45 | 21.86 | 15.71 | 14.05 | 18.84 | 14.56 | 20.52 | 17.58 |
| LCD | 15.08 | 14.50 | 20.73 | 17.67 | 15.31 | 13.14 | 15.77 | 15.87 | 17.93 | 16.22 |
| SCD | 19.42 | 18.69 | 24.64 | 23.20 | 19.22 | 16.57 | 18.90 | 20.10 | 19.42 | 20.02 |
| HUB | 66.46 | 70.63 | 62.38 | 78.16 | 103.10 | 71.15 | 75.88 | 47.48 | 56.42 | 70.18 |
| LMB | 32.93 | 27.17 | 34.67 | 29.44 | 28.30 | 24.27 | 26.24 | 21.26 | 27.50 | 27.98 |
| NLB | 53.60 | 47.84 | 42.23 | 52.79 | 43.78 | 43.65 | 49.35 | 47.59 | 51.16 | 48.00 |
| NLBY | 34.92 | 30.99 | 35.99 | 52.31 | 42.23 | 40.45 | 42.25 | 39.90 | 34.21 | 39.25 |
| LN | 11.70 | 15.46 | 13.97 | 16.78 | 14.29 | 9.98 | 13.97 | 13.85 | 17.55 | 14.17 |
| WN | 15.32 | 17.28 | 18.76 | 23.24 | 16.68 | 15.51 | 26.88 | 22.31 | 23.24 | 19.91 |
| LLS | 14.63 | 19.92 | 20.19 | 27.49 | 16.65 | 21.06 | 26.73 | 17.74 | 22.53 | 20.77 |
| NFW | 16.20 | 18.11 | 20.13 | 25.72 | 38.33 | 17.57 | 24.31 | 20.47 | 19.99 | 22.31 |
| Mean | 26.80 | 26.78 | 28.40 | 32.70 | 31.22 | 25.57 | 30.08 | 25.04 | 27.93 | 28.28 |

Population abbreviations and phenotypic trait abbreviations are shown in Table 1 and Figure 2, respectively.

### 3.4. Principal Component Analysis of Phenotypic Traits

Through the principal component analysis of twelve phenotypic traits, three principal components with eigen values greater than one was obtained, and the cumulative contribution rate reached 74.40%. On the PC1 (38.983%) and PC2 (25.262%) planes, three naturally segregating groups were formed. The first group is mainly characterized by plant

shape, including plant height (PH), diameter breast height (DBH), long crown diameter (LCD), short crown diameter (SCD), and length of the main branch of the year (LMB). The second group is mainly characterized by needle trait, comprising length of needles (LN), width of needles (WN), needles fascicles width (NFW), and length of leaf sheath (LLS). The third group mainly represents characteristics of lateral branch traits, including the number of lateral branches (NLB), and the number of lateral branches of the year (NLBY) was shown (Figure 3).

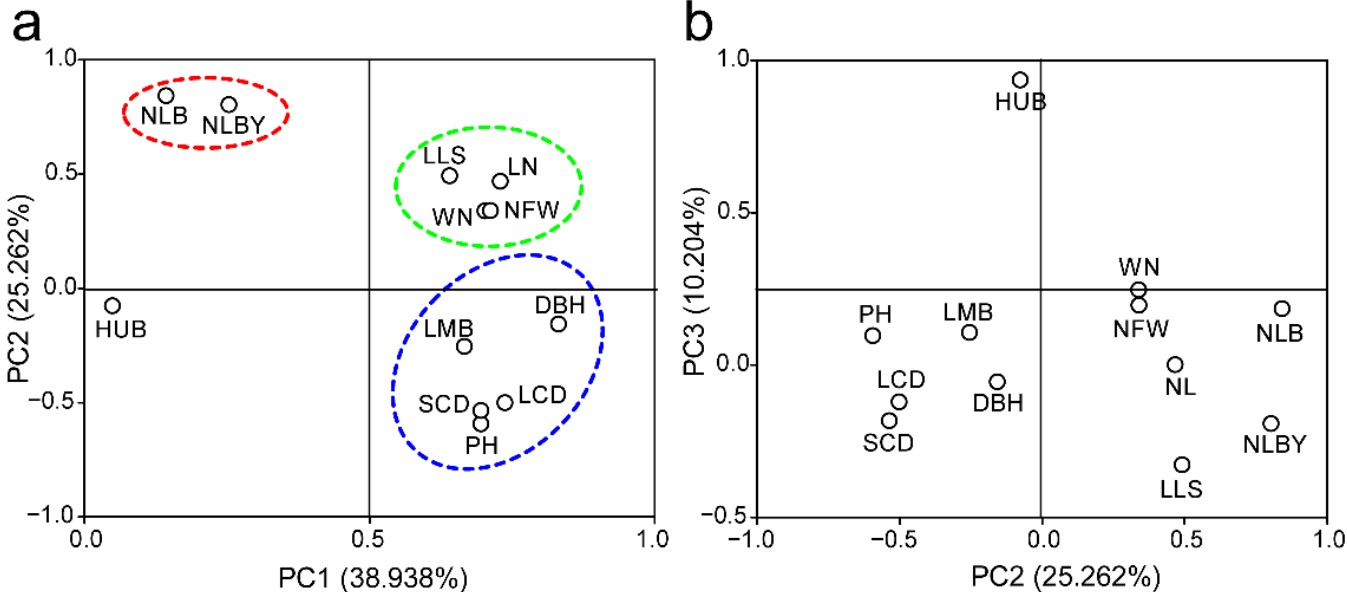

**Figure 3.** Principal component analysis of phenotypic traits of *P. yunnanensis* populations. Drawing a scatterplot with PC1 as the *X*-axis and PC2 as the *Y*-axis, three naturally segregating groups were formed (**a**). Draw a scatterplot with PC2 as the *X*-axis and PC3 as the *Y*-axis (**b**). Phenotypic trait abbreviations are shown in Figure 2.

### 3.5. Correlation between Phenotypic Traits and Spatial Arrangement and Ecological Factors

The correlation analysis among twelve phenotypic traits was performed (Figure 4a). There were significantly positive correlations among the traits that are mainly involved in forestry production, such as plant height, diameter breast height, long crown diameter, and short crown diameter. This demonstrated that the selected superior family can have both rapid growth and excellent plant shape, which is in line with the breeding goal. The correlation analysis between twelve phenotypic traits and spatial arrangement and ecological factors shown (Figure 4b) were summarized as follows: LON (1.61) > WP (0.76) > MAT (0.45) > MAP (0.41) > DT (0.17) > DP (−0.20) > WT (−0.27) > LAT (−0.84) > EL (−1.76). Since the main traits related to forestry production are PH and DBH, the spatial arrangement and ecological factors that are significantly correlated to plant height and diameter breast height are mainly discussed here. Among them, mean annual temperature, mean temperature of driest quarter, mean temperature of the wettest quarter is significantly positively correlated with diameter breast height ($p < 0.05$), and latitude is significantly negatively correlated with diameter breast height ($p < 0.05$), while there is no correlation between plant height and various spatial arrangement and ecological factors, implying that temperature is the key factor restricting the radial growth of *P. yunnanensis*.

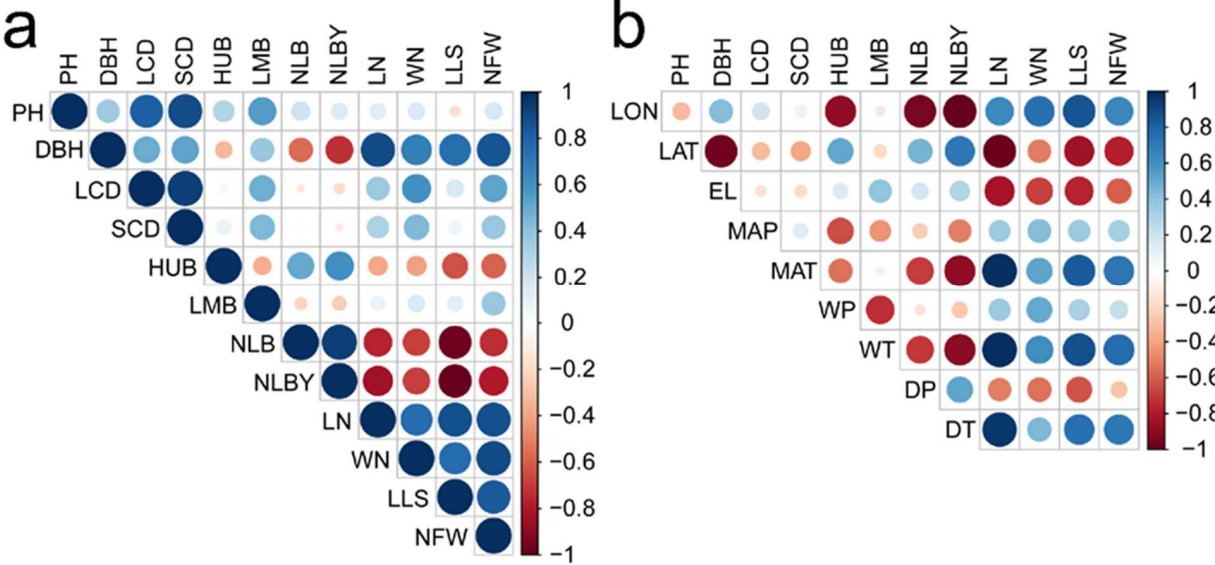

**Figure 4.** Correlation analysis among phenotypic traits (**a**) and spatial arrangement and ecological factors (**b**) of *P. yunnanensis*. Blue represents positive correlation while maroon represents negative correlation. The darker the color, the stronger the correlation. Phenotypic trait abbreviations are shown in Figure 2.

### 3.6. Cluster Analysis of P. yunanensis Population

Based on the twelve phenotypic traits, nine populations were divided into three clusters by UPGMA as shown (Figure 5). The first cluster was composed of SB, XP, HZ, XC, TC, CY, and CH populations, while the second cluster included the LF population and the third cluster contained the YR population. Combined with the geographical coordinates of each sampling point, it was founded that the phenotypic characteristics are not clustered strictly according to geographical distance, implying that there is a discontinuity in the variation of phenotypic traits of *P. yunnanensis*.

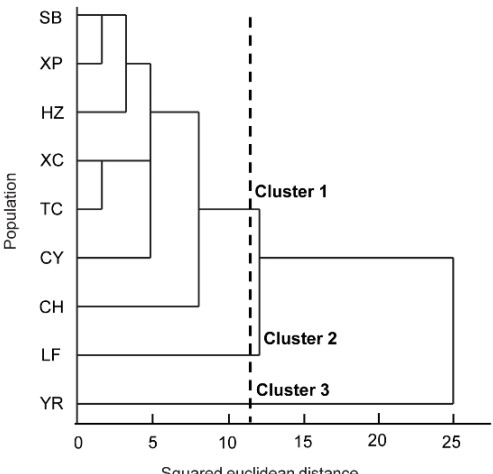

**Figure 5.** Cluster analysis based on the phenotypic traits of *P. yunnanensis* populations. Table 1 indicated the number of populations. Population abbreviations are shown in Table 1.

### 3.7. Superior Family Selection of P. yunnanensis

By analysis of the average value, range, standard deviation, and the basic value of nine populations (Table 4), the scoring standard for each trait was established (Table S2), thus, the total score of each family was formed. The *t*-test was performed on the total scores of all families and the score values of all families were normally distributed. One hundred

and twenty-eight families were primarily selected from 258 families with a selection rate of 49.61%. The comprehensive evaluation is based on the scores of each trait. The more traits with higher scores, the better the overall performance of the single plants is. The heat map shown (Figure 6) was used to screen the superior families in the primary selection, and twelve superior families were selected with a selection rate of 10%. The codes were CY24, HZ1, HZ2, SB9, SB10, SB18, SB25, LF7, CH1, CH7, LF17, and LF18, respectively. The numbers after the abbreviations of different populations represent the different families.

**Table 4.** Growth performance and form quality performance of each population of *P. yunnanensis*.

| Traits | Mean | Range | SD | Basic Value | CV (%) |
|--------|------|-------|-----|-------------|--------|
| PH | 100.20 | 126.82 | 21.52 | 121.72 | 21.48 |
| DBH | 46.26 | 93.10 | 8.46 | 54.72 | 18.29 |
| LCD | 119.45 | 87.33 | 15.21 | 134.66 | 12.73 |
| SCD | 98.22 | 121.00 | 15.55 | 113.77 | 15.83 |
| HUB | 8.83 | 56.33 | 6.64 | 15.47 | 75.20 |
| LMB | 52.55 | 85.67 | 12.04 | 64.59 | 22.91 |
| NLB | 0.82 | 0.90 | 0.12 | 0.94 | 14.63 |
| NLBY | 0.89 | 0.82 | 0.09 | 0.98 | 10.11 |
| LN | 27.21 | 36.85 | 4.65 | 31.86 | 17.09 |
| WN | 1.08 | 0.97 | 0.20 | 1.28 | 18.52 |
| LLS | 13.10 | 19.01 | 3.02 | 16.12 | 23.05 |
| NFW | 2.05 | 3.97 | 0.40 | 2.45 | 19.51 |

Phenotypic trait abbreviations are shown in Figure 2.

Based on the results aforementioned, three principal components with accumulative contribution rates exceeding 70% were obtained from the principal component analysis to calculate their weights, 0.523, 0.340, and 0.137, respectively. After weighted calculation, families are sorted according to the comprehensive score of the principal components, and the score ranges from −1.507 to 1.419. Using 10% as the selection rate of superior families, the top 25 families were selected with a range from 0.958 to 1.419. The codes of the selected superior families and the scores of each principal component were listed (Table S3).

For the superior families selected by the comprehensive scoring method (Table S4), the actual gain range of each trait is from 0.04% to 33.50%, of which the height under the branches is the highest and the number of lateral branches is the lowest. The average actual gain of the twelve phenotypic traits is 16.44%. For the superior families selected by principal component analysis (Table S4), the actual gain range of each trait is 1.22% to 24.66% in which the height under the branches is the highest and the number of lateral branches of the year is the lowest. The average actual gain of the twelve phenotypic traits is 16.26%. Comparing the selection results of the two methods, it can be concluded that the average actual gain of the twelve phenotypic traits obtained by the comprehensive scoring method is higher than that from the principal component analysis. The height under the branches among the twelve phenotypic traits gets the highest actual gains based on two selection methods.

Comparing the superior families selected by the two methods (Table 5), there were eleven superior families that overlapped, and the selection rate was 4.26% (Figure 7). The actual gain range of each trait of the selected superior families is from 0.02% to 32.23%. In each phenotypic trait, the height under the branches has the highest actual gain, whereas the number of lateral branches and the number of lateral branches of the year was the lowest. The average actual gain of the twelve phenotypic traits reached 16.66%, which is better than the effect of the two methods used alone. Among the eleven superior families selected, the majority belonged to the LF and SB populations (three families, respectively), followed by the HZ and CH populations (two families, respectively).

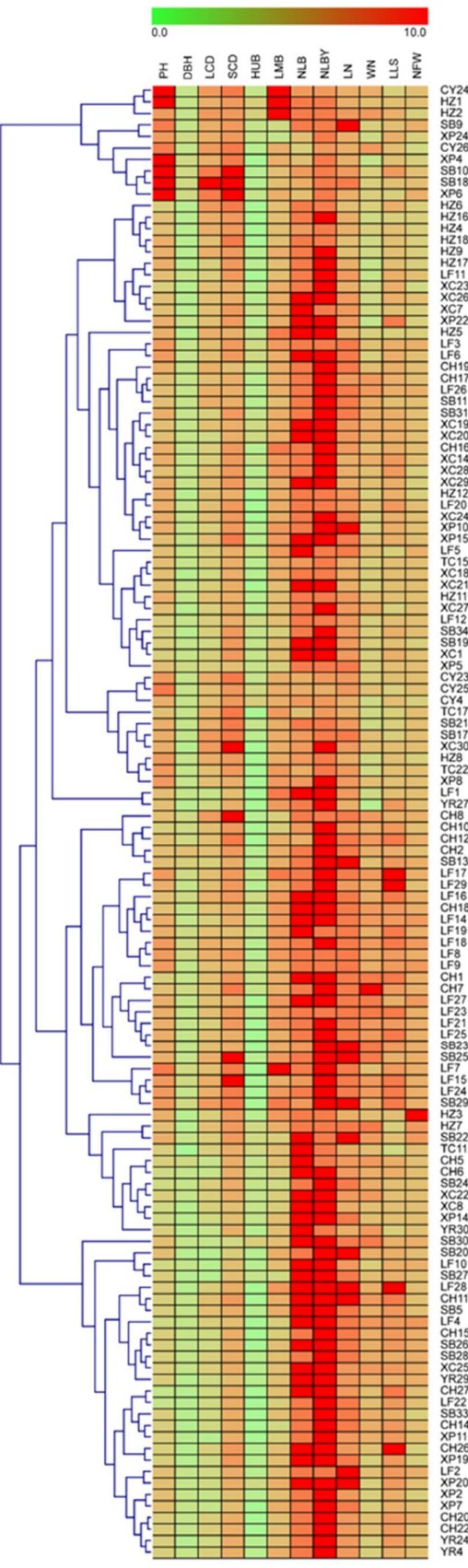

**Figure 6.** Superior families of *P. yunnanensis* evaluated by the heatmap. Population abbreviations and phenotypic trait abbreviations are shown in Table 1 and Figure 2, respectively.

**Table 5.** The growth, form quality performance, and realized gains of superior families of *P. yunnanensis*.

| Code | PH | DBH | LCD | SCD | HUB | LMB | NLB | NLBY | LN | WN | LLS | NFW |
|---|---|---|---|---|---|---|---|---|---|---|---|---|
| CY24 | 156.00 | 51.05 | 145.33 | 121.33 | 14.33 | 83.67 | 0.74 | 0.76 | 26.83 | 1.33 | 11.37 | 2.39 |
| HZ1 | 157.67 | 49.61 | 139.67 | 108.67 | 27.33 | 84.33 | 0.87 | 0.90 | 25.70 | 1.20 | 12.16 | 2.28 |
| HZ2 | 133.67 | 51.38 | 145.67 | 113.67 | 24.33 | 90.33 | 0.86 | 0.82 | 30.00 | 1.39 | 14.92 | 1.94 |
| SB10 | 151.67 | 59.03 | 159.67 | 134.33 | 7.33 | 54.67 | 0.73 | 0.86 | 27.87 | 1.12 | 16.38 | 2.16 |
| SB18 | 164.33 | 64.70 | 171.00 | 141.00 | 10.33 | 55.00 | 0.67 | 0.83 | 30.77 | 1.13 | 11.95 | 2.13 |
| SB25 | 121.00 | 54.95 | 144.67 | 129.33 | 8.33 | 68.67 | 0.82 | 0.94 | 35.13 | 1.48 | 15.39 | 2.58 |
| LF7 | 142.33 | 60.12 | 139.00 | 117.33 | 1.67 | 90.67 | 0.81 | 0.94 | 33.50 | 1.22 | 16.24 | 2.52 |
| CH1 | 98.00 | 52.13 | 129.67 | 109.67 | 13.33 | 57.67 | 0.93 | 0.93 | 33.17 | 1.46 | 17.96 | 2.78 |
| CH7 | 113.00 | 56.20 | 135.33 | 120.67 | 16.33 | 57.67 | 0.86 | 0.92 | 33.90 | 1.59 | 16.08 | 2.63 |
| LF17 | 133.00 | 57.02 | 135.33 | 106.00 | 12.00 | 75.67 | 0.87 | 0.94 | 32.57 | 1.42 | 19.95 | 2.53 |
| LF18 | 129.67 | 57.21 | 141.00 | 97.00 | 8.00 | 72.33 | 0.87 | 0.92 | 33.73 | 1.42 | 17.17 | 2.84 |
| Mean | 136.39 | 55.76 | 144.21 | 118.90 | 13.03 | 71.88 | 0.82 | 0.89 | 31.20 | 1.34 | 15.42 | 2.43 |
| ΔG% | 26.54 | 17.04 | 17.17 | 16.83 | 32.23 | 26.89 | 0.02 | 0.02 | 12.78 | 19.53 | 15.02 | 15.81 |

Population abbreviations and phenotypic trait abbreviations are shown in Table 1 and Figure 2, respectively. The numbers after the abbreviations of different populations represent the different families.

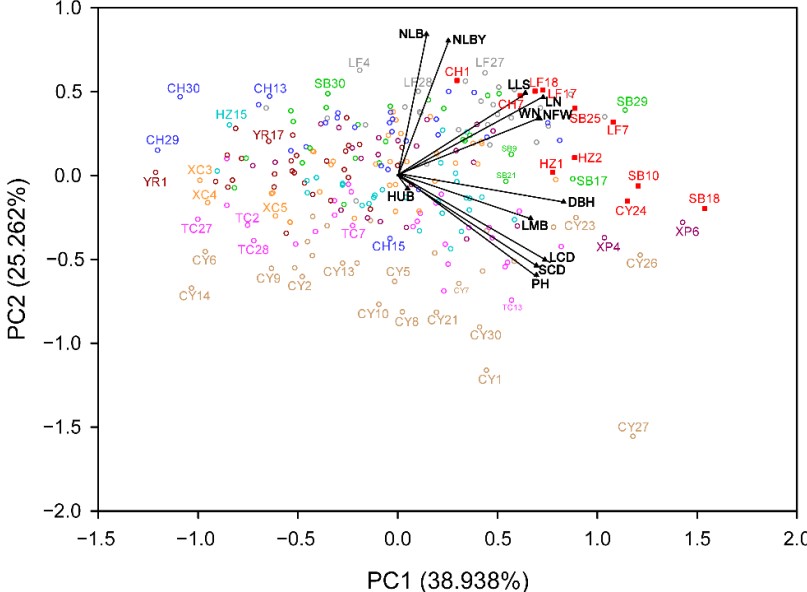

**Figure 7.** Ordination in the space of the first two principal component axes based on twelve phenotypic traits of 258 families of *P. yunnanensis*. The black solid triangle represents twelve different phenotypic traits. The red solid box represents the superior family selected while the black circle represents other families. Different populations were colored differently. Population abbreviations and phenotypic trait abbreviations are shown in Table 1 and Figure 2, respectively.

Moreover, we evaluated the relationship among the phenotypic traits under the study by family × trait biplots. The family × trait biplot displayed a large proportion (64.2%) of the information in the data obtained from the families (Figure 7). Orthogonal vectors in this plot represent that the corresponding traits vary independently, while a deviation orthogonally indicates correlation (positive if the angle is acute, negative if the angle is obtuse). For example, the acute angles between diameter breast height, length of the main branch of the year, long crown diameter, short crown diameter, and plant height showed that these traits are positively and significantly correlated. The trait profiles of the families are visualized by their respective distances from the biplot origin. The expression amount of each family relative to a certain trait is easily identified from the biplots based on their distance from the biplot origin and closeness to the trait vertex. Thus, we found that families that had high values for diameter breast height were also with high values of length of needles and needles fascicles width.

## 4. Discussion

### 4.1. Sources of Phenotypic Variation in Different Populations

Through the study of twelve phenotypic traits of nine natural populations of *P. yunnanensis* in the entire distribution area, it is found that the phenotypic diversity is abundant among and within populations, which is similar to the results reported by [29] on Saharan maize (*Zea mays* L.). The phenotypic differentiation coefficient is an objective reflection of the adaptability of plants to the environment. The average phenotypic differentiation coefficient of *P. yunnanensis* population in this study was 91.23%, which is much higher than that of *P. albicaulis* (15%) [30], *Populus simonii* (47.1%) [31], *Quercus robur* (10% to 15%) [32], *Juglans regia* (41.07%) [15], *Cunninghamia lanceolata* (34.54%) [33], *Orchis mascula* (20%) [34], and *Rosa platyacantha* (16.51%) [35]. The nine populations have obvious regionalized distribution characteristics, resulting in rich phenotypic variation among populations. The phenotypic variation among populations is higher than that within a population, which is consistent with the results of *Tetracentron sinense* [36] and *Lonicera caerulea* [37]. Therefore, we concluded that the phenotypic variation of *P. yunnanensis* mainly comes among populations which possibly are related to the geographical isolation of the various populations, which hindered the pollination of wind-pollinated plants and the gene exchange between populations, and increased the differentiation among populations. Variation among populations reflects differences in geographic and reproductive isolation, and it is an important manifestation of intraspecific diversity. Based on the sequencing data of mtDNA and cpDNA of *P. yunnanensis* [38], indicated that geographical and environmental factors together created stronger and more discrete genetic differentiation than isolation by distance alone, and disclosed the importance of ecological factors in forming or maintaining genetic divergence across a complex landscape. In addition, 20 natural populations collected from the main distribution areas of *P. yunnanensis* were analyzed by SSR marker, demonstrating that there was no evident sign of geographical isolation, and the ecological isolation was more obvious than geographical isolation, whereas the genetic differentiation among populations may be influenced by geographical, climatic, or soil factors [39]. Considering the limitations of the sampling range in the previous study, we extended the research object to the whole distribution area of *P. yunnanensis*. Our phenotypic data support the view that the discrete genetic differentiation was collectively shaped by geographical and environmental factors.

### 4.2. Phenotypic Variation Characteristics of Populations

The coefficient of variation is a quantitative description of the degree of dispersion of traits and is positively correlated with phenotypic diversity. The order of the average coefficient of variation was: lateral branches (38.41%) > plant shape (29.39%) > needles (19.29%). The coefficient of variation of branch traits is twice that of needle traits. According to the research on Norway spruce (*Picea abies* (L.) Karst.), the coefficient of variation of the crown is significantly higher than that of the main stem [40]. The coefficient of variation of stem on *Acacia auriculiformis* A. Cunn. Ex. Benth. (8.3% to 13.1%) is also smaller than that of branches (24.6% to 32.6%) [41]. This is possible due to the reason that the branching status can reflect the growth status of the plant to some extent [28]. Therefore, different populations may adapt to diverse environments by adjusting the number or length of the lateral branches.

### 4.3. Correlation between Phenotypic Traits and Spatial Arrangement and Ecological Factors

In the correlation analysis, we mainly focused on the traits that were closely related to forestry production, such as plant height, diameter breast height, long crown diameter, and short crown diameter. These traits own a significant positive correlation, demonstrating that superior families with both fast growth and excellent plant shape can be screened when family selection is carried out, which is in line with our breeding goal. The number of lateral branches and the number of lateral branches of the year has a weak correlation with other traits, indicating that branch traits are relatively independent. The nine populations

are distributed in southwestern China, with latitudes ranging from 24.07° to 28.62° N and longitude ranging from 97.35° to 105.93° E. The differences in these geographical environmental factors are the external reasons for the phenotypic diversity. In this study, except elevation, mean annual precipitation, mean precipitation of wettest quarter, and mean precipitation of driest quarter had no significant effect on the phenotypic traits; the other five factors had significant effects on the phenotypic traits. This is in stark contrast to the research that the Pedunculate Oak (*Quercus robur* L.) is very sensitive to drought and humidity [42]. Since *P. yunnanensis* is the main timber plant and afforestation tree species in southwestern China, two indicators, plant height, and diameter breast height are mainly concerned in forestry production. As shown (Figure 4b), the correlation between plant height and spatial arrangement, and ecological factors were not significant. Thus, only spatial arrangement and ecological factors that were significantly correlated to diameter breast height were discussed here. Among them, latitude is negatively correlated with diameter breast height. The mean temperature of the driest quarter, mean annual temperature, and mean temperature of the wettest quarter are positively correlated with diameter breast height, indicating that temperature is the main ecological factor affecting the diameter breast height. The study related to the radial growth reported by Shen et al. (2020) also showed that temperature has the most important influence on the growth of *P. yunnanensis* [43]. The length of needles of *P. yunnanensis* is significantly correlated with almost all environmental factors [44]. Variations in plant morphological characteristics usually have adaptive significance. The law of interaction between plant populations and ecological factors is relatively complicated, while different species often have different behaviors. For example, the radial growth of *Salzmann pine* has different correlations with ecological factors at different research sites. The southern site is positively correlated with spring precipitation, and the northern site is correlated with temperature [45]. The radial growth of *P. nigra* is strongly positively correlated with precipitation in summer, late summer, and early autumn, and weakly negatively correlated with temperature in spring and summer [46]. The radial growth of *P. contorta* is positively correlated with precipitation at low altitudes and is positively correlated with the temperature at high altitudes [47]. The breast diameter of *Larix olgensis* is significantly positively correlated with longitude and precipitation [48].

Through principal component analysis and cluster analysis, the nine natural populations can be divided into three clusters. The first cluster consists of SB, XP, HZ, XC, TC, CY, and CH populations, which grow in wet areas with high annual precipitation in Yunnan, Sichuan, Guizhou, and Tibet provinces. The second cluster is the LF population, which grows in the middle latitudes of Yunnan Province. The third cluster is the YR population, which grows in the northcentral region of Yunnan Province. By comparing the phenotypic traits of the three clusters, among the twelve traits investigated, the measured values of nine traits (plant height, diameter breast height, long crown diameter, short crown diameter, length of the main branch of the year, length of needles, width of needles, length of leaf sheath, and needles fascicles width) of LF were significantly higher than those of other populations, so that the LF population was considered to be a population with obvious advantages. The measured value of four phenotypic traits (plant height, long crown diameter, short crown diameter, and the number of lateral branches) of the YR population showed obvious disadvantages among nine populations. The various populations are not clustered strictly according to geographic distance, and similar conclusions were obtained in the Xu et al. (2016) study on *P. yunnanensis* [49] and similar to the variation of *P. roxburghii* [50]. It was shown that the variation of phenotypic traits of the *P. yunnanensis* population is discontinued, which may be due to the fact that different populations grow in different habitats. Due to the barriers between the populations, it is difficult for genes to be spread among populations. It is difficult to carry out gene exchange even among populations that are geographically close, which increases the chance of independent differentiation of various populations. In addition, it was found that the similarity of growth environment may be a key factor for the geographical clustering of *P. pinaster* and *Crataegus azarolus*

populations [51,52], whereas the growth environment of *P. yunnanensis* populations is mostly relatively independent, and local microclimates are complex and changeable. Environmental differences have also exacerbated the independent differentiation between various populations. Therefore, nine natural populations of *P. yunnanensis* are not clustered according to geographic distance.

*4.4. Superior Family Selection*

Abundant variations among various traits are the basis for superior family selection. Based on the investigation of twelve traits of 258 families of *P. yunnanensis*, we found that it has rich variations of traits from various sources and contains great potential for selection. It is a common method to calculate the comprehensive score and then sort superior single plants by principal component analysis [53,54]. In this study, the cumulative contribution rate of the traits we focused on for forestry production reached 74.40%. Taking advantage of these traits, a total of 25 superior families were selected with a selection rate of 10%, and a higher actual gain was obtained [55]. However, Verardi et al. verified the results of joint selection by calculating genetic gain. The comprehensive scoring method can avoid the omission of superior single plants due to the significant differences in the selection indicators between single plants. Comparing the selection results of the two methods, it was found that one of the superior families is different, indicating that the superior family with higher levels of various traits can be avoided by the heat map evaluation.

The advantage of principal component analysis is that it simplifies the selection procedure, while a comprehensive scoring method can grasp the comprehensive trait performance of a single plant. Using a single method to select the superior family cannot give consideration to all the traits. Therefore, we combined principal component analysis and comprehensive scoring methods and found those eleven selected superior families overlapped. Among the eleven selected superior families, six families were from SB and LF in central Yunnan Province, indicating that *P. yunnanensis* is most suitable for growth locally. Among the superior families selected in this experiment, the actual gain of traits closely related to forestry production is higher, which is helpful to provide excellent germplasm materials for the genetic and breeding research of *P. yunnanensis*.

**5. Conclusions**

In order to tap more potential breeding materials, the phenotypic diversity of provenance and family experimental forest was evaluated. Abundant phenotypic variation was revealed both within and between populations. Among the twelve populations, the phenotypic variation mainly comes from the variation among populations. Temperature is the most important environmental factor affecting the diameter of breast height. The clustering analysis showed that the variations of phenotypic traits are discontinuous and that the differentiation of various populations is relatively independent. The rich phenotypic variations are the basis for the selection of superior families. Combined with the comprehensive scoring method and principal component analysis, eleven superior families with 16.66% average actual gain were successfully obtained. This study provides a number of solid breeding materials and technical support for the directional cultivation and popularization of the excellent industrial raw material tree species.

**Supplementary Materials:** The following supporting information can be downloaded at: https://www.mdpi.com/article/10.3390/f13040618/s1, Table S1: Measurement method and the type of twelve traits.; Table S2: Comprehensive scoring assessment criteria for each trait of *P. yunnanensis*; Table S3: The principal component scores and ranking of superior individuals of *P. yunnanensis*. Table S4: The mean values and realized gains for growth and form quality traits of families of *P. yunnanensis* based on comprehensive scoring and principal component analysis.

**Author Contributions:** K.C. designed the research; Z.L. and J.L. conducted the research; Y.M. and C.G. analyzed the data; Z.L. wrote the paper. All authors have read and agreed to the published version of the manuscript.

**Funding:** This work was funded by the Essential Scientific Research of Chinese National Non-profit Institute (CAFYBB2019ZB006), Yunnan Applied Basic Research Projects (2019FA013), National Natural Science Foundation of China (32022058), and Training Objects of Technological Innovation Talents in Yunnan Province (2019HB074).

**Institutional Review Board Statement:** Not applicable.

**Data Availability Statement:** Not applicable.

**Conflicts of Interest:** The authors declare that they have no conflict of interest.

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
