# Peer review of "Phenotypic Diversity Analysis and Superior Family Selection of Industrial Raw Material Forest Species-Pinus yunnanensis Franch"

_forests, doi:10.3390/f13040618_

Round 1

Reviewer 1 Report

I feel that the abstract of the paper should be rewritten ( I send copy of suggested abstract prepared by me). The language of revised text must be evaluated by an expert of English.

Author Response

Dear editors:

Thanks for your kind assistance to our manuscript entitled “Phenotypic variation and superior family selection of industrial raw material forest species-Pinus yunnanensis (forests-1619158)”. We have revised our manuscript according to your suggestions. You will find the detailed revision (in red color) as follows.

Comments from Academic editor for Authors: Although the manuscript is more a technical report than a scientific paper, it could merit to go on peer review process, but first it must be extensively reviewed for a minimum quality of the scientific English. Not only in formal aspects (widespread grammatical and typing errors) but also in the scientific terminology (land diameter instead of basal diameter....and many others). This makes the reading and further review really difficult, and will indeed affect negatively the recommendation by the reviewers.

Response: A native English speakers helped us to revise the language of the manuscript.

Response to REVIEWER #1

1.Comment: I feel that the abstract of the paper should be rewritten (I send copy of suggested abstract prepared by me). The language of revised text must be evaluated by an expert of English.

Response: We have revised it.

You are welcome to give other suggestions to us.

Sincerely,

Zirui Liu,

State Key Laboratory of Tree Genetics and Breeding

Institute of Highland Forest Science, Chinese Academy of Forestry, Kunming 650233, PR China

Fax: +86-871-63860027. E-mail address:   [email protected]

Reviewer 2 Report

GENERAL COMMENTS

The revised paper is in scope with the MDPI Forests journal scope. Generally, provenience trial researches are a hot topic in forestry. Paper analyzed twelve phenotypic characteristics of five-year-old Pinus yunnanensis seedlings taken from nine populations, which is the most dominant conifer in the analyzed area.

Data set and research design are on a high level. On the other hand, lots of mistakes are noted, which must be improved, and this paper must be pushed up to level if authors want to publish them in the Q1 journal. Also, English is at a very low level and must be improved. In some parts, the manuscript is very hard to follow.

My main concern is about the author’s interpretation of “family level”. The authors don’t analyze the population’s diversity on the molecular level, and cannot observe each tree as a new family. That is not true, and this is the main structural weakness of this manuscript which must be changed to the next rounds of review. Therefore, I recommended EiC major review which required structural changes and English language improvement. I am open to reviewing this significantly improved manuscript.

SPECIFIC COMMENTS

AIM

L What are „geographic ecological factors“? Maybe spatial arrangement and ecological factors??

I can not find what is the theoretical background of this research.

L 100 The fine varieties is an incorrect term. Please find a better word

M&M

Please change rephrase subtitle 2.1. to „Site description and stand characteristics “.

You don’t analyze 258 families, your sample obtained the above-mentioned number of analyzed trees, this is a big difference.

Also, your title must be changed.

L113 please, add a new paragraph, and provide more information about provenience train setup. This is important for your work.

Fig 1. The site abbreviations on the map are too small. Also in the whole paper, you use a term population for your sites, only in Fig. 1 do you use the Sample point.

L 133 Dou you mean on five-year-old seedlings (or trees) when said, five-year-old family? This is a completely wrong term.

In this sub-section, all abbreviations of analyzed tree characteristics should be added. Also, the number of crown and diameter measurements per tree is missed here.

L 166-176 Except on Fig. 5, which is completely hard to read and too complex to provide a deeper conclusion, in the whole paper only inter and intra-population variability were analyzed. Likewise, your data set is not suitable for following half- and full-sib lines. Based on your data you can select only superior tree specimens, but this is out of the main hypothesis of your paper. Hence this whole sub-section 2.5. is not justified for use.

L 182-187 A lot of sentences from this paragraph are for the Results section.

RESULTS

L 195 Please, all basic statistic abbreviations include in M&M and in following chapters please use only their abbreviations.

Table 3 is hard to read. I wonder it is a better option to present this data graphically with a bar chart with post hoc test and SD?

The title of Table 4. is superficial, please provide a more informative title. Also, I cannot understand what is it exactly.

Fig 2 Title must be more informative. You don’t need to introduce abbreviations in each table of figure titles.

Once again, I don’t know what is geographic ecological factors. I heard the first time for this.

Fig. 2 It is too small letters.

Fig 3. Change trait abbreviation from red to black color and use a bigger font.

You don’t select superior families. You can not call each tree family!

Table 5. On this table, you present population statistics, not each family….

Tables 6, 7, and 8 should be moved to the Supplementary material.

Fig. & is completely non-informative. I suggest you use different colors for the population.

The discussion section is generally well written. After reading the whole section, only sub-section 4.4. should be updated. In this section, lots of results are repeated. Also, I suggest authors read this paper about different phenologicaly varieties' responses to drought and their diversity (https://doi.org/10.3390/f12070930).

Based on the required changes, the conclusion section should be improved.

Author Response

Dear editors:

Thanks for your kind assistance to our manuscript entitled “Phenotypic variation and superior family selection of industrial raw material forest species-Pinus yunnanensis (forests-1619158)”. We have revised our manuscript according to your suggestions. You will find the detailed revision (in red color) as follows.

Comments from Academic editor for Authors: Although the manuscript is more a technical report than a scientific paper, it could merit to go on peer review process, but first it must be extensively reviewed for a minimum quality of the scientific English. Not only in formal aspects (widespread grammatical and typing errors) but also in the scientific terminology (land diameter instead of basal diameter....and many others). This makes the reading and further review really difficult, and will indeed affect negatively the recommendation by the reviewers.

Response: A native English speakers helped us to revise the language of the manuscript.

Response to REVIEWER #2

1. Comment: L What are “geographic ecological factors”? Maybe spatial arrangement and ecological factors??

Response: We have modified “geographic ecological factors” to “spatial arrangement and ecological factors”.

2. Comment: I cannot find what is the theoretical background of this research.
Response: Added the theoretical background in instruction.

3. Comment: L 100 The fine varieties is an incorrect term. Please find a better word.
Response: We have modified “fine varieties” to “superior varieties”.

4. Comment: Please change rephrase subtitle 2.1. to “Site description and stand characteristics ”.
Response: We have revised it.

5. Comment: You don’t analyze 258 families, your sample obtained the above-mentioned number of analyzed trees, this is a big difference.
Response: In Section 2.1, we introduced in detail how we established the provenance and family experimental forest. 258 superior trees from nine provenances were collected and seeds were harvested in single plant. Thirty seedlings originated from each superior tree were cultivated. Thus, 258 families were formed. The random block design composed by single plant plot with 30 blocks was adopted. For provenance and family experimental forest, each provenance is randomly arranged within each block, and each family is randomly arranged within the provenance. In this study, every tree in the experimental forest was investigated. Therefore, we actually analyzed 258 families.

6. Comment: Also, your title must be changed.

Response: We kept the original “Phenotypic variation and superior family selection of industrial raw material forest species-Pinus yunnanensis”.

7. Comment: L113 please, add a new paragraph, and provide more information about provenience train setup. This is important for your work.
Response: A new paragraph was added to introduce how we setup the provenance and family experimental forest.

8. Comment: Fig 1. The site abbreviations on the map are too small. Also in the whole paper, you use a term population for your sites, only in Fig. 1 do you use the Sample point.
Response: We have increased the type size in Figure 1 and modified “sample points” to “sampling population”.

9. Comment: L 133 Dou you mean on five-year-old seedlings (or trees) when said, five-year-old family? This is a completely wrong term.
Response: We have modified family to seedlings.

10. Comment: In this sub-section, all abbreviations of analyzed tree characteristics should be added. Also, the number of crown and diameter measurements per tree is missed here.
Response: All abbreviations of analyzed tree characteristics was added in 2.2. The related traits of crown and diameter of each tree were measured three times respectively, and the measurement method and trait type were shown (Table 2).

11.Comment: L 166-176 Except on Fig. 5, which is completely hard to read and too complex to provide a deeper conclusion, in the whole paper only inter and intra-population variability were analyzed. Likewise, your data set is not suitable for following half- and full-sib lines. Based on your data you can select only superior tree specimens, but this is out of the main hypothesis of your paper. Hence this whole sub-section 2.5. is not justified for use.
Response: The heat map in Figure 5 is the second step of the comprehensive scoring method to select superior families. According to the color distribution, we can get more excellent families, compared to families with scores in the top 1/10.

In Section 2.1, we introduced in detail how we established the provenance and family experimental forest. 258 superior trees from nine provenances were collected and seeds were harvested in single plant. Thirty seedlings originated from each superior tree were cultivated. Thus, 258 families were formed. The random block design composed by single plant plot with 30 blocks was adopted. For provenance and family experimental forest, each provenance is randomly arranged within each block, and each family is randomly arranged within the provenance. In this study, every tree in the experimental forest was investigated. Therefore, we actually analyzed 258 families.

12. Comment: L 182-187 A lot of sentences from this paragraph are for the Results section.
Response: We have moved this paragraph to results section.

13. Comment: L 195 Please, all basic statistic abbreviations include in M&M and in following chapters please use only their abbreviations.

Response: Modified.

14. Comment: Table 3 is hard to read. I wonder it is a better option to present this data graphically with a bar chart with post hoc test and SD?

Response: Changed to bar chart.

15. Comment: The title of Table 4. is superficial, please provide a more informative title. Also, I cannot understand what is it exactly.

Response: We have modified the title of Table 4 to “Characteristic of phenotypic traits in populations of P. yunnanensis revealed by variance components.”

The coefficient of variation is used to compare the degree of dispersion of phenotypic traits. The larger the coefficient of variation, the higher the dispersion of representative traits. The greater the degree of phenotypic variability.

16. Comment: Fig 2 Title must be more informative. You don’t need to introduce abbreviations in each table of figure titles.

Response: Modified.

17. Comment: Once again, I don’t know what is geographic ecological factors. I heard the first time for this.

Response: Modified “geographic ecological factors” to “spatial arrangement and ecological factors”.

18. Comment: Fig. 2 It is too small letters.

Response: Modified.

19. Comment: Fig 3. Change trait abbreviation from red to black color and use a bigger font.

Response: Modified.  

20. Comment: You don’t select superior families. You cannot call each tree family!

Response: In Section 2.1, we introduced in detail how we established the provenance and family experimental forest. 258 superior trees from nine provenances were collected and seeds were harvested in single plant. Thirty seedlings originated from each superior tree were cultivated. Thus, 258 families were formed. The random block design composed by single plant plot with 30 blocks was adopted. For provenance and family experimental forest, each provenance is randomly arranged within each block, and each family is randomly arranged within the provenance. In this study, every tree in the experimental forest was investigated. Therefore, we actually analyzed 258 families.

 21. Comment: Table 5. On this table, your present population statistics, not each family….

Response: We have modified family to population.

22. Comment: Tables 6, 7, and 8 should be moved to the Supplementary material.

Response: We have moved Tables 6, 7 and 8 to the Supplementary material.

23. Comment: Fig. & is completely non-informative. I suggest you use different colors for the population.

Response: We have colored the different populations differently.

24. Comment: The discussion section is generally well written. After reading the whole section, only sub-section 4.4. should be updated. In this section, lots of results are repeated. Also, I suggest authors read this paper about different phenological varieties' responses to drought and their diversity (https://doi.org/10.3390/f12070930).

Response: We have updated the sub-section 4.4 and the repeated results has been deleted. The above-mentioned article has been read and cited.

25. Comment: Based on the required changes, the conclusion section should be improved.

Response: Modified.

You are welcome to give other suggestions to us.

Sincerely,

Zirui Liu,

State Key Laboratory of Tree Genetics and Breeding

Institute of Highland Forest Science, Chinese Academy of Forestry, Kunming 650233, PR China

Fax: +86-871-63860027. E-mail address:   [email protected]

Reviewer 3 Report

Dear authors,

I want to congratulate you for this exciting work with significant contributions to guide the selection of superior populations of Pinus yunnanensis based on phenotypic traits variation analysis and population scoring.  

Line 62. I highly recommend changing the citation format without starting the sentence with [13]. 

Line 66. It could be interesting to explain why you select these phenotypic characteristics, why the traits analyzed are important, and its interest in breeding programs. 

Line 88. Wrong citation format due to the lack of the year between parentheses.

Line 89. The selection of qualitative indicators is highly undesirable for that kind of study. Why do not select quantitative traits?

Line 115. Punctuation error in (Figure. 1)

Figure 1. Very low resolution. The quality of the figure must be improved to a minimum resolution of 300 ppp.

Table 1. The order of the abbreviations described under the table must be in the same order as the columns in the table.

Line 132. A more detailed description of how the measurement of phenotypic traits was desirable.

Line 195. Extra space after parentheses.

Table 2 and 3. Table 3 must be presented before table 2. It is recommended to decrease the font size, leave space between numbers and symbols and avoid dividing words and numbers between lines in the same row (pop-ulations, er-or, trai-t). Rotate the table if necessary. In table 3, the order of the abbreviations described under the table must be in the same order as the columns in the table. The explanation of using different letters after numbers in the same column in table 3 must be further explained. The units of each phenotypic trait must be indicated.

Table 4. The order of the abbreviations described under the table must be in the same order as the columns in the table. The units must be indicated.

Line 250. Include the contribution of each principal component in the text.

Figure 2. Include in the PCA the vectors representing each phenotypic trait the easily interpret the correlation with the principal components and the distribution of the samples in clusters. This way of representing a PCA is a little confusing. The resolution of the figure is very low, and hardly could read the labels. A minimum resolution of 300 PPP is recommended.

Figures 3 and 4. The resolution of the figure is very low, and hardly could read the labels. A minimum resolution of 300 PPP is recommended.

Line 292. Why LF population has obvious advantages? This sentence does not fit well in the results of cluster analysis. Describe in more detail this affirmation in the discussion. 

Line 307. This sentence fits better in Methods.

Line 314. Punctuation error in Figure. 5.

Line 316. Punctuation error in commas. A more detailed explanation about the index included in the new abbreviations of the populations must be done.

Figure 5. The resolution of the figure is very low, and hardly could read the labels. A minimum resolution of 300 ppp is recommended. The order of the abbreviations described under the figure must be in the same order as the columns in the heat map. Grammatical error in the description of the figure Heatmap.

Table 5. Avoid dividing tables between pages. Include the units. The order of the abbreviations described under the table must be in the same order as the rows in the table.

Table 6. Include the units. The order of the abbreviations described under the table must be in the same order as the rows in the table. It is recommended to decrease the font size, leave space between numbers and symbols and avoid dividing numbers between lines in the same row. Rotate the table if necessary.

Table 9. The description of the table does not fit with the information included. A properly description of the data shown must be included. The order of the abbreviations described under the table must be in the same order as the columns in the table.

Figure 6. The resolution of the figure is very low, and hardly could read the labels. A minimum resolution of 300 PPP is recommended. 

Line 446. Punctuation error.

Line 472. Wrong citation format.

Thanks for taking into consideration my recommendations.

Best regards.

Author Response

Dear editors:

Thanks for your kind assistance to our manuscript entitled “Phenotypic variation and superior family selection of industrial raw material forest species-Pinus yunnanensis (forests-1619158)”. We have revised our manuscript according to your suggestions. You will find the detailed revision (in red color) as follows.

Comments from Academic editor for Authors: Although the manuscript is more a technical report than a scientific paper, it could merit to go on peer review process, but first it must be extensively reviewed for a minimum quality of the scientific English. Not only in formal aspects (widespread grammatical and typing errors) but also in the scientific terminology (land diameter instead of basal diameter....and many others). This makes the reading and further review really difficult, and will indeed affect negatively the recommendation by the reviewers.

Response: A native English speakers helped us to revise the language of the manuscript.

Response to REVIEWER #3

1. Comment: Line 62. I highly recommend changing the citation format without starting the sentence with [13].

Response: We have modified [13] to the end of the sentence.

2. Comment: Line 66. It could be interesting to explain why you select these phenotypic characteristics, why the traits analyzed are important, and its interest in breeding programs.

Response: Added “These phenotypic traits selected were mainly interested in forestry production, and were also easily accessible, and guided for forestry production.”

3. Comment: Line 88. Wrong citation format due to the lack of the year between parentheses.
Response: Modified.

4. Comment: Line 89. The selection of qualitative indicators is highly undesirable for that kind of study. Why do not select quantitative traits?
Response: We strongly agree with you. The predecessors established a tree selection system by evaluating qualitative traits. But we believe that the inability to quantify these qualitative indicators is a big drawback. Therefore, all quantitative traits were selected in our study.

5. Comment: Line 115. Punctuation error in (Figure. 1)
Response: Modified.

6. Comment: Figure 1. Very low resolution. The quality of the figure must be improved to a minimum resolution of 300 ppp.
Response: Modified.

7. Comment: Table 1. The order of the abbreviations described under the table must be in the same order as the columns in the table.
Response: Modified.

8. Comment: Line 132. A more detailed description of how the measurement of phenotypic traits was desirable.
Response: Modified.

9. Comment: Line 195. Extra space after parentheses.
Response: Modified.

10. Comment: Table 2 and 3. Table 3 must be presented before table 2. It is recommended to decrease the font size, leave space between numbers and symbols and avoid dividing words and numbers between lines in the same row (populations, eror, trait). Rotate the table if necessary. In table 3, the order of the abbreviations described under the table must be in the same order as the columns in the table. The explanation of using different letters after numbers in the same column in table 3 must be further explained. The units of each phenotypic trait must be indicated.
Response: Modified. Table 3 was changed to bar chart, and before table 2.

11. Comment: Table 4. The order of the abbreviations described under the table must be in the same order as the columns in the table. The units must be indicated.
Response: Modified.

12. Comment: Line 250. Include the contribution of each principal component in the text.
Response: Modified.

13. Comment: Figure 2. Include in the PCA the vectors representing each phenotypic trait the easily interpret the correlation with the principal components and the distribution of the samples in clusters. This way of representing a PCA is a little confusing. The resolution of the figure is very low, and hardly could read the labels. A minimum resolution of 300 PPP is recommended.

Response: Modified.

14. Comment: Figures 3 and 4. The resolution of the figure is very low, and hardly could read the labels. A minimum resolution of 300 PPP is recommended.

Response: Modified.

15. Comment: Line 292. Why LF population has obvious advantages? This sentence does not fit well in the results of cluster analysis. Describe in more detail this affirmation in the discussion.

Response: Added. By comparing the phenotypic traits of the three clusters, among the twelve traits investigated, the measured values of nine traits (plant height, land diameter, long crown diameter, short crown diameter, length of the main branch of the year, length of needles, width of needles, length of leaf sheath and needles fascicles width) of LF were significantly higher than those of other populations, so that the LF population was considered to be a population with obvious advantages. The measured value of four phenotypic traits (plant height, long crown diameter, short crown diameter and number of lateral branches) of YR population showed obvious disadvantages among nine populations.

16. Comment: Line 307. This sentence fits better in Methods.

Response: This sentence has moved to section 2.5.

17. Comment: Line 314. Punctuation error in Figure. 5.

Response: Modified.

18. Comment: Line 316. Punctuation error in commas. A more detailed explanation about the index included in the new abbreviations of the populations must be done.

Response: Modified.

19. Comment: Figure 5. The resolution of the figure is very low, and hardly could read the labels. A minimum resolution of 300 ppp is recommended. The order of the abbreviations described under the figure must be in the same order as the columns in the heat map. Grammatical error in the description of the figure Heatmap.

Response: Modified.

20. Comment: Table 5. Avoid dividing tables between pages. Include the units. The order of the abbreviations described under the table must be in the same order as the rows in the table.

Response: Modified.

21. Comment: Table 6. Include the units. The order of the abbreviations described under the table must be in the same order as the rows in the table. It is recommended to decrease the font size, leave space between numbers and symbols and avoid dividing numbers between lines in the same row. Rotate the table if necessary.

Response: Modified.

22. Comment: Table 9. The description of the table does not fit with the information included. A properly description of the data shown must be included. The order of the abbreviations described under the table must be in the same order as the columns in the table.

Response: Modified.

23. Comment: Figure 6. The resolution of the figure is very low, and hardly could read the labels. A minimum resolution of 300 PPP is recommended.

Response: Modified.

24. Comment: Line 446. Punctuation error.

Response: Modified.

25. Comment: Line 472. Wrong citation format.

Response: Modified.

You are welcome to give other suggestions to us.

Sincerely,

Zirui Liu,

State Key Laboratory of Tree Genetics and Breeding

Institute of Highland Forest Science, Chinese Academy of Forestry, Kunming 650233, PR China

Fax: +86-871-63860027. E-mail address:   [email protected]

Reviewer 4 Report

The manuscript is well presented and sounds good. Here are my comments Figures quality needs to be improved for high resolution. Section 2.2 Provide a table of the 12 traits and describe how these traits were collected as well as their nature (qualitative or quantitative). This section "Standardized of data" need. to come before you start the analysis. depending on the nature of the data the standardization will not be the same for all the traits please revise this.

Author Response

Dear editors:

Thanks for your kind assistance to our manuscript entitled “Phenotypic variation and superior family selection of industrial raw material forest species-Pinus yunnanensis (forests-1619158)”. We have revised our manuscript according to your suggestions. You will find the detailed revision (in red color) as follows.

Comments from Academic editor for Authors: Although the manuscript is more a technical report than a scientific paper, it could merit to go on peer review process, but first it must be extensively reviewed for a minimum quality of the scientific English. Not only in formal aspects (widespread grammatical and typing errors) but also in the scientific terminology (land diameter instead of basal diameter....and many others). This makes the reading and further review really difficult, and will indeed affect negatively the recommendation by the reviewers.

Response: A native English speakers helped us to revise the language of the manuscript.

Response to REVIEWER #4

1.Comment: The manuscript is well presented and sounds good. Here are my comments Figures quality needs to be improved for high resolution. Section 2.2 Provide a table of the 12 traits and describe how these traits were collected as well as their nature (qualitative or quantitative). This section "Standardized of data" need to come before you start the analysis depending on the nature of the data the standardization will not be the same for all the traits, please revise this.

Response: We have improved all the figures quality to a higher resolution.

Table 2 was added to describe the measurement method and the type of twelve traits.

We have reversed the order of section 2.3 (Data analysis) and section 2.4 (Standardized of data).

You are welcome to give other suggestions to us.

Sincerely,

Zirui Liu,

State Key Laboratory of Tree Genetics and Breeding

Institute of Highland Forest Science, Chinese Academy of Forestry, Kunming 650233, PR China

Fax: +86-871-63860027. E-mail address:   [email protected]

Round 2

Reviewer 2 Report

The revised manuscript was significantly improved, but I have a few structural and important concerns. 
Firstly, the manuscript title was very poorly written and non-informative for readers. Firstly, the founder’s name must be added after the Latin name. Likewise, this paper was not focused on the selection of superior families, they focused on their diversity and superior traits important for foresters. Then, authors repeat words in the title and in KW, this is not necessary. 
Also, the LSD test results in Fig 2 are strange and this must be checked before further manuscript processing. If LSD follows deviations among the population, they are completely wrong and require authors' explanation and in-depth check. 
Finally, I recommended EiC one more review of this paper with a focus on the above-mentioned important questions.

Author Response

Dear editors:

Thanks for your kind assistance to our manuscript entitled “Phenotypic variation and superior family selection of industrial raw material forest species-Pinus yunnanensis (forests-1619158)”. We have revised our manuscript according to your suggestions. You will find the detailed revision (in red color) as follows.

Response to REVIEWER #2

1.Comment: Firstly, the manuscript title was very poorly written and non-informative for readers. Firstly, the founder’s name must be added after the Latin name. Likewise, this paper was not focused on the selection of superior families, they focused on their diversity and superior traits important for foresters. Then, authors repeat words in the title and in KW, this is not necessary.

Also, the LSD test results in Fig 2 are strange and this must be checked before further manuscript processing. If LSD follows deviations among the population, they are completely wrong and require authors' explanation and in-depth check.

Response: We have changed the title to “Phenotypic diversity analysis and superior family selection of industrial raw material forest species-Pinus yunnanensis Franch.”

The repeated words in key words were modified.

In the last revision, multiple comparisons were made between 12 traits per population by mistake. Figure 2 has now been changed to multiple comparisons across nine populations for each trait.

Thanks a lot for your reminder! Please forgive us for our carelessness.

You are welcome to give other suggestions to us.

Sincerely,

Zirui Liu,

State Key Laboratory of Tree Genetics and Breeding

Institute of Highland Forest Science, Chinese Academy of Forestry, Kunming 650233, PR China

Fax: +86-871-63860027. E-mail address:   [email protected]
